# Machine learning for predicting cognitive deficits using auditory and demographic factors

**Christopher E. Niemczak**[1,2‡*], **Basile Montagnese**[1‡], **Joshua Levy**[3,4,5,6], **Abigail M. Fellows**[2], **Jiang Gui**[6,7], **Samantha M. Leigh**[2], **Albert Magohe**[8], **Enica R. Massawe**[8], **Jay C. Buckey**[1,2]

**1** Geisel School of Medicine at Dartmouth, Space Medicine Innovations Laboratory, Lebanon, NH, United States of America, **2** Dartmouth Health, Department of Medicine, Division of Hyperbaric Medicine, Lebanon, NH, United States of America, **3** Dartmouth Health, Department of Pathology and Laboratory Medicine, Lebanon, NH, United States of America, **4** Dartmouth Health, Department of Dermatology, Lebanon, NH, United States of America, **5** Geisel School of Medicine at Dartmouth, Epidemiology, Lebanon, NH, United States of America, **6** Geisel School of Medicine at Dartmouth, Program in Quantitative Biomedical Sciences, Lebanon, NH, United States of America, **7** Geisel School of Medicine at Dartmouth, Biomedical Data Science, Lebanon, NH, United States of America, **8** Muhimbili University of Health and Allied Sciences, Dar es Salaam, Tanzania

‡ CEN and BM are co-first authorship on this work.
* Christopher.e.niemczak@dartmouth.edu

**Data Availability Statement:** Data cannot be shared publicly because of PHI (i.e., data contain potentially identifying or sensitive patient information) and data use agreement. Data are

## Abstract

### Importance

Predicting neurocognitive deficits using complex auditory assessments could change how cognitive dysfunction is identified, and monitored over time. Detecting cognitive impairment in people living with HIV (PLWH) is important for early intervention, especially in low- to middle-income countries where most cases exist. Auditory tests relate to neurocognitive test results, but the incremental predictive capability beyond demographic factors is unknown.

### Objective

Use machine learning to predict neurocognitive deficits, using auditory tests and demographic factors.

### Setting

The Infectious Disease Center in Dar es Salaam, Tanzania

### Participants

Participants were 939 Tanzanian individuals from Dar es Salaam living with and without HIV who were part of a longitudinal study. Patients who had only one visit, a positive history of ear drainage, concussion, significant noise or chemical exposure, neurological disease, mental illness, or exposure to ototoxic antibiotics (e.g., gentamycin), or chemotherapy were excluded. This provided 478 participants (349 PLWH, 129 HIV-negative). Participant data were randomized to training and test sets for machine learning.

available from the Dartmouth Institutional Data Access / Ethics Committee for researchers who meet the criteria for access to confidential data. Contact information for a data access Dartmouth's Committee for the Protection of Human Subjects (cphs@dartmouth.edu).

**Funding:** YES. This study was funded by the National Institutes of Health (NIH), grant number 5R01DC009972 to principal investigator Jay C. Buckey M.D. The content of this report is solely the responsibility of the authors and does not necessarily represent the official views of the NIH.

**Competing interests:** NO authors have competing interests.

## Main outcome(s) and measure(s)

The main outcome was whether auditory variables combined with relevant demographic variables could predict neurocognitive dysfunction (defined as a score of <26 on the Kiswahili Montreal Cognitive Assessment) better than demographic factors alone. The performance of predictive machine learning algorithms was primarily evaluated using the area under the receiver operational characteristic curve. Secondary metrics for evaluation included F1 scores, accuracies, and the Youden's indices for the algorithms.

## Results

The percentage of individuals with cognitive deficits was 36.2% (139 PLWH and 34 HIV-negative). The Gaussian and kernel naïve Bayes classifiers were the most predictive algorithms for neurocognitive impairment. Algorithms trained with auditory variables had average area under the curve values of 0.91 and 0.87, F1 scores (metric for precision and recall) of 0.81 and 0.76, and average accuracies of 86.3% and 81.9% respectively. Algorithms trained without auditory variables as features were statistically worse (p < .001) in both the primary measure of area under the curve (0.82/0.78) and the secondary measure of accuracy (72.3%/74.5%) for the Gaussian and kernel algorithms respectively.

## Conclusions and relevance

Auditory variables improved the prediction of cognitive function. Since auditory tests are easy-to-administer and often naturalistic tasks, they may offer objective measures or predictors of neurocognitive performance suitable for many global settings. Further research and development into using machine learning algorithms for predicting cognitive outcomes should be pursued.

## 1. Introduction

Complex auditory tasks, such as perceiving, understanding, and responding to speech in background noise engage a variety of neurocognitive domains [1]. Our previous results show a relationship between complex auditory tasks (termed: central auditory tasks) and cognitive function [2, 3]. We have shown a moderate correlation between the Montreal Cognitive Assessment (MoCA), a validated cognitive screening measure, and performance on speech-in-noise tasks in Tanzania (R2 = .14, p < .001) [3] and China (r2 = 0.30, p < .001) [4]. These relationships have also been found with other neurocognitive assessments, such as the Cogstate battery and the Test of Variables of Attention [2, 3]. In Alzheimer's disease, Gates et al. found a relationship between central auditory performance (i.e., dichotic tests) and the Cognitive Ability Screening Instrument [5]. These analyses suggest that by combining central auditory tests with other simple measures, predicting neurocognitive function could be improved. Furthermore, recent studies demonstrate links between peripheral auditory ability and neurocognitive function [6]. Peripheral auditory tests often require periods of concentrated effort with varying auditory inputs, potentially engaging diverse neurocognitive functions. Auditory tests may add to factors such as age, education, socioeconomic status, and disease severity for screening for or even predicting of neurocognitive function. This could have a profound impact, especially in low- to middle-income countries, where access to formal neurocognitive

assessment is often limited due to a shortage of trained administrators and the high costs associated with testing and where lack of education may impair performance on simpler screening tests.

Recent HIV treatment advances have reduced HIV related mortality drastically [7–9]. Nevertheless, even with modern combination antiretroviral therapy (cART), cognitive impairment (HIV associated neurocognitive disorder) still occurs, especially in low- middle-income countries [10–14]. While cART has reduced HIV-related dementia substantially [7], varying levels of neurocognitive dysfunction persist in people living with HIV (PLWH), even with suppressed viral loads [15]. Identifying, tracking, and potentially predicting neurocognitive dysfunction in this population would provide crucial benefits for PLWH. Biomarkers for brain health and techniques to identify cognitive dysfunction are important for reducing mortality, morbidity, and disease progression, not only in PLWH but for other neurocognitive disorders such as Alzheimer's and Parkinson's disease [7].

Machine learning (ML) can identify patterns and associations that characterize the relationship between diverse clinical variables (auditory tests and demographic variables in this case) and a target of interest (cognitive performance). Such patterns are typically impossible to detect manually as they reflect nuanced medical information that complicate data interpretation. These algorithms, when properly trained, can make inferences on previously unseen patient cohorts and can supplement medical decision making [16, 17]. Additionally, the use of machine learning in healthcare has the potential to revolutionize personalized medicine. By analyzing vast amounts of patient data, ML algorithms can identify unique patterns and subgroups within populations, allowing for more precise and tailored treatment approaches for individuals. This level of customization can lead to improved patient outcomes and a more efficient allocation of healthcare resources in low- middle-income countries.

The goal of this exploratory study was to use ML to predict neurocognitive dysfunction in adults living with and without HIV using central and peripheral auditory tests as well as demographic and clinical factors such as age, education, and CD4 count. By using an ML approach with auditory variables, we hoped to improve the prediction of neurocognitive function, and leverage this technique to identify, track, and predict neurocognitive deficits in PLWH. The relationship between auditory processing measures and cognition likely relates to the similarities in brain functions used for advanced auditory processing and higher order cognition. Therefore, we hypothesized that machine learning using auditory test variables could improve cognitive impairment prediction beyond demographic and clinical factors. If auditory function combined with other easily measured factors is a successful proxy or complementary assessment for neurocognitive function, these tests could initiate new diagnostic, treatment, and monitoring techniques for neurocognitive function in HIV and other neurocognitive diseases all over the world.

## 2. Methods

### 2.1. Data collection

**Subjects.**   Adults were tested at the Infectious Disease Center in Dar es Salaam, Tanzania as part of an ongoing longitudinal research study. Enrollees were tested every 6 months, and some have been coming for up to 5 years. Recruitment for this study started on July 7th 2017 and ended on November 20th 2022. The Institutional Review Boards of both Dartmouth College and Muhimbili University of Health and Allied Sciences approved this study's research protocol. All research was completed in accordance with the Helsinki Declaration. All participants provided written informed consent to participate.

To ensure accuracy of analysis and control for confounding auditory variables we imposed exclusion criteria. Analyzed subjects had to have more than one visit with complete central auditory test data. This excluded the greatest number of participants (excluded n = 345). Furthermore, individuals were excluded if they had hearing loss (>25 dB HL) on their pure tone audiogram at any frequency from 0.5 to 4 kHz (excluded n = 37) or if they had abnormal middle ear function (Tympanogram Type B or C–excluded 79). Thus, from the initial 939 screened individuals, only 478 participants were incorporated into the ML analysis (***Supplementary Fig 1 in** S1 File)*. Table 1 shows the demographics of the sample population.

**Auditory data collection.** Audiological testing was performed as previously described using a hearing assessment system built by Creare, LLC (Niemczak et al., 2021) [11]. Audiometric measures of peripheral hearing sensitivity using a Békésy tracking technique were used to calculate pure tone averages (PTA) of 0.5, 1.0, 2.0, and 4.0 kHz. Central auditory processing in Swahili, the native language of all participants, was evaluated using four measures: an adaptive Gap Detection test (Gap), the Triple Digit test (TDT), the Hearing in Noise Test (HINT), and the Staggered Spondaic Words (SSW) test. The test battery combined three domains of central auditory function: temporal auditory processing (Gap), speech-in-noise ability (TDT and HINT), and dichotic auditory processing (SSW). For this paper the term '*auditory variables*' includes both peripheral and central auditory performance metrics, unless otherwise stated.

**Cognitive data collection.** Cognitive function was also measured as described previously (Niemczak et al., 2021). A score <26 on the Kiswahili version of the Montreal Cognitive Assessment (MoCA) was used to indicate cognitive impairment [18–20]. While the MoCA is a cognitive screening tool, and not a full neurocognitive assessment, it provides an overall measure of cognition on domains of visuospatial and executive functioning, attention, concentration, memory, language, and delayed recall [18]. The MoCA is commonly used to screen for cognitive impairments, particularly in detecting mild cognitive impairment (MCI) and early signs of dementia. It provides a quick and reliable assessment of multiple cognitive domains and is widely used in clinical and research settings to detect cognitive changes [16].

## 2.2. Data preprocessing

In clinical applications, the most likely approach to using hearing tests to track cognition would be to measure them repeatedly and assess if results are worsening over time. To include this possibility in the prediction, longitudinal data on auditory were included for each subject. Trajectories of all auditory measures over multiple visits were plotted and mean and slope values for each subject were calculated. Before calculating each mean and slope value, an analytical regression line was calculated using each auditory test plotted over time and outliers >2 standardized residuals from the regression line were removed to eliminate distortion due to

**Table 1. Demographics.**

|  |  | PLWH | HIV-Negative | P-Value |
|---|---|---|---|---|
| **Number of Participants** |  | 349 | 129 | N/A |
| **Age** (SD) |  | 39.9 (14.9) | 28.8 (12.1) | < .001 |
| **Male** (% of cohort) |  | 92 (26.4%) | 63 (48.8%) | < .001 |
| **Years of Education** (SD) |  | 8.63 (2.87) | 9.7 (3.1) | < .001 |
| **Number of Visits** (SD) |  | 7.5 (2.5) | 5.85 (2.3) | < .001 |
| **CD4 Count** (SD) |  | 652.9 (267.9) | 824.8 (262.8) | < .001 |
| **Cognitive Deficit on MoCA** (% of cohort) |  | 139 (39.8%) | 34 (26.4%) | < .001 |

outliers. This longitudinal approach also reduced variation from using measurements from a single experimental session.

## 2.3. Data analysis

**Feature selection.** Feature variables relevant for predicting cognitive impairment were derived from both the subjects' audiological test scores (central and peripheral) and their demographic information. Features were initially chosen based on previous literature which demonstrated correlations between central auditory variables and cognitive function [2, 3, 21, 22]. Additionally, feature ranking algorithms were performed using the Diagnostic Feature Designer in MATLAB® 2021a. The absolute value two-sample T-test with pooled variance estimate as well as the Bhattacharyya algorithm, which ranks continuous features by the minimum predicted attainable classification error (Chernoff bound), were used to select features suspected to have predictive capabilities in a ML model. In addition to initial feature rankings, models were trained, validated, and tested iteratively with different subsets of the remaining features (i.e., manual selection) to determine a final set of features (***Supplementary Fig 2 in S1 File***). Since HIV status was both discrete and binary it could not be ranked in the same method; however, from our manual feature selection scheme, previous literature, and overall goal the work in Tanzania, we decided to incorporate HIV status as a predictive feature. Overall, the 17 selected features chosen for the development of our ML algorithm consisted of: 1) HIV status, 2) age, 3) education years, 4–5) mean and slope of the TDT score, 6–7) mean and slope of the HINT score, 8–9) mean and slope of the Gap threshold score, 10–11) the mean and slope of the SSW (labeled as Score), 12–15) mean and slope of the PTA score (i.e. peripheral auditory function) for both left and right ears, and 16–17) mean and slope of the CD4 count. Furthermore, to evaluate the specific impact of the auditory variables, ML algorithms were trained and tested with the features determined above both with and without all auditory variables. This provided a quantifiable justification for including auditory variables in the ML models.

**Data preparation.** To prepare datasets for training and testing classification models, the overall dataset was randomly separated into a training/validation set and a held-out test set (n = 382 and 96, respectively). The training/validation and test sets were checked to ensure similar percentages of cognitively impaired individuals in both sets (37% and 33% respectively) which was representative of the overall population between the partitioned sets. The training/ validation set was used to train the model and provide an estimate of the model's capabilities while tuning hyperparameters which control the learning process of an algorithm, such as the types of distributions a model uses to learn. The test set was a mutually exclusive dataset used to evaluate the model's performance on a new, untrained data set.

## 2.4. Developing the algorithm

**Model construction.** The ML algorithm was developed using the classification learner app on MATLAB® 2021a. The target variable (impairment on MoCA at the most recent visit) and chosen features were input from the partitioned training/validation data set and different classification algorithms were simultaneously trained and validated using 10-fold cross-validation. The available algorithms that were trained/validated on the data included decision trees, logistic regression classifiers, naïve Bayes classifiers, support vector machines, ensemble classifiers (i.e., LogitBoost model), and neural network classifiers. After the initial area under the curve (AUC) evaluation of the different classifiers (***Table 2)*** the naïve Bayes classifiers were the most promising and chosen for further investigation. The two specific naïve Bayes classifiers available were the Gaussian naïve Bayes classifier and the kernel naïve Bayes classifier which

**Table 2. Summary of AUC values from different classification algorithms.**

| Algorithm Type | AUC _With_ Auditory Variables (SD) | AUC _Without_ Auditory Variables (SD) |
|---|---|---|
| _Decision Trees_ | 0.73 (0.02) | 0.72 (0.06) |
| _Logistic Regression_ | 0.78 (0.06) | 0.79 (0.05) |
| _Support Vector Machines_ | 0.81 (0.01) | 0.79 (0.03) |
| _Ensemble_ | 0.81 (0.04) | 0.75 (0.04) |
| _Neural Network_ | 0.70 (0.07) | 0.71 (0.03) |
| _Gaussian Naïve Bayes_ | 0.91 (0.04) | 0.82 (0.05) |
| _Kernel Naïve Bayes_ | 0.87 (0.05) | 0.78 (0.05) |

assume the likelihood for each predictor to be modeled as Gaussian distributions or a kernel density estimation respectively. Since naïve Bayes algorithms derive from Bayes's theorem, a significant assumption of class-conditional independence exists (i.e., the features are independent) [23, 24]. The optimal model configuration was selected via a simplified MATLAB hyperparameter sweep (**Supplementary Table 1 in S1 File**).

**Model evaluation.** Classification algorithm performance was evaluated using the area under the receiver operating characteristic curve (AUC). The AUC value offers a collective measure of an algorithm's performance across the complete collection of classification thresholds and is usually considered the most important metric [25]. Confidence intervals for the AUC values were constructed through 10 iterations of randomizing the data partitions and training/testing the algorithms. Secondary metrics for evaluation include F1 scores of the algorithms, accuracies of the algorithms, and the Youden's indices for the algorithms. The F1 score represents the geometric mean between precision and recall and can take on a value between 0 to 1, where 1 is perfect precision/recall. While the AUC is calculated at thresholds between the true positive rate and the false positive rate, the F1 score is a straightforward calculation involving the overall recall and precision of the model, which yields performance information with respect to false negatives and false positives in one number. Youden's index captures the optimal trade-off between sensitivity and specificity and is selected via a sensitivity analysis to determine the optimal decision threshold to inform other classification metrics (e.g., F1-score) and provide clinically relevant results [26]. To assess the AUC and accuracy between Gaussian and Kernel Naïve Bayes p-values were calculated using a two-proportion z-test for accuracy, and the DeLong method [27] for AUC.

**Model interpretation.** Developed naïve Bayes models were evaluated to determine the most impactful features for predicting cognitive performance. Shapley values [28] were calculated via MATLAB functions. Shapley values provide global feature importance rankings for any machine learning models after fitting the model based on predictive capacity. They help define which features contributed most to the ML algorithm prediction output for any given patient.

## 3 Results

### 3.1. Performance of the classification models

The performance of classification models showed strong results with and without auditory variables. The difference in ROC curves highlighted the differing capabilities of the algorithms tested and that the predictive capabilities well surpassed the chance threshold (i.e., prediction at random) (_Fig 1_). The mean AUC values for the Gaussian and kernel naïve Bayes classifiers

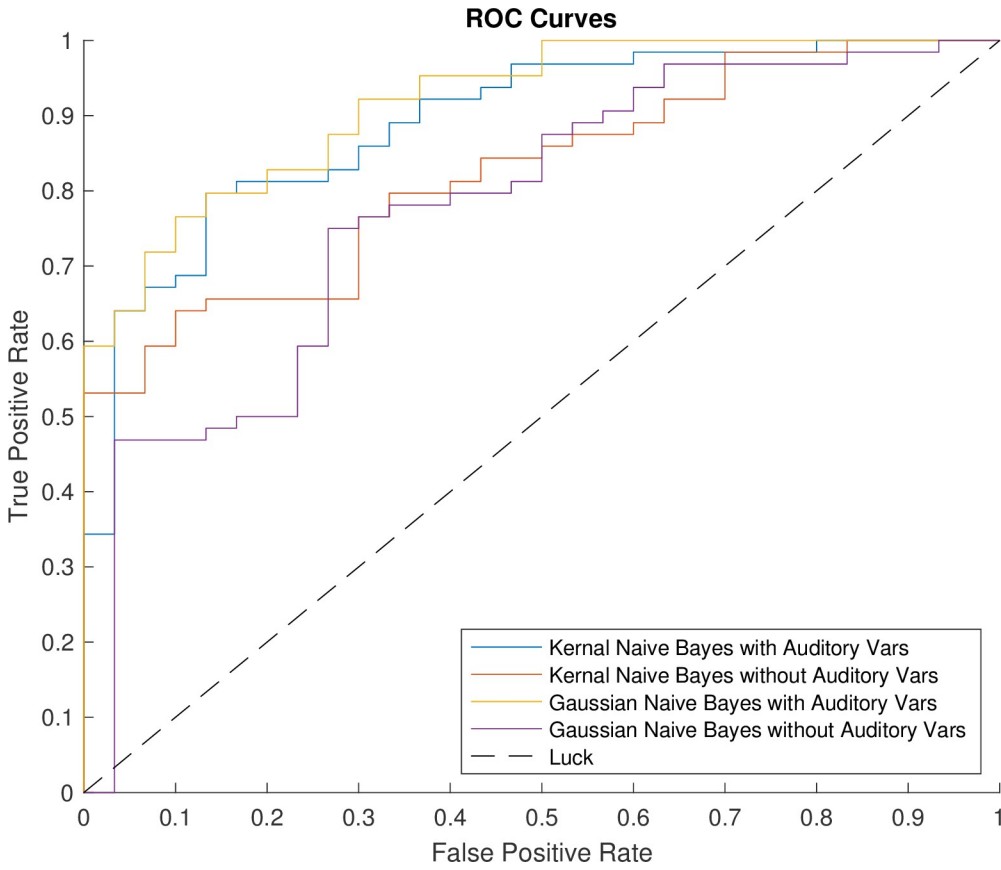

**Fig 1. Mean test ROC curves.**

including auditory variables were 0.91 and 0.87 respectively. These high AUC values are complimented by high accuracies on the held-out test sets as shown in **Tables 2 & 3**. Furthermore, the secondary metrics for evaluation such as the F1-score, accuracy, and Youden's index are summarized in **Table 3**. Test confusion matrices for the Gaussian and Kernel Naïve Bayes classifiers with and without auditory variables are shown in **Figs 2 and 3**. The results demonstrate significant difference between the algorithms with and without auditory variables incorporated.

**Table 3. Summary of results of naïve Bayes algorithms.**

|  | **With Auditory Variables** | **Without Auditory Variables** | **p-value** |
|---|:---:|:---:|:---:|
| | *Gaussian Naïve Bayes* | | |
| **AUC** (SD) | 0.91 (0.04) | 0.82 (0.05) | < .001 |
| **F1** | 0.81 | 0.66 | |
| **Accuracy** (SD) | 86.3% (3.4) | 72.3% (1.9) | < .001 |
| **Youden's Index** | 0.72 | 0.54 | |
| | *Kernel Naïve Bayes* | | |
| **AUC** (SD) | 0.87 (0.05) | 0.78 (0.05) | < .001 |
| **F1** | 0.76 | 0.64 | |
| **Accuracy** (SD) | 81.9% (3.5) | 74.5% (4.3) | < .001 |
| **Youden's Index** | 0.66 | 0.48 | |

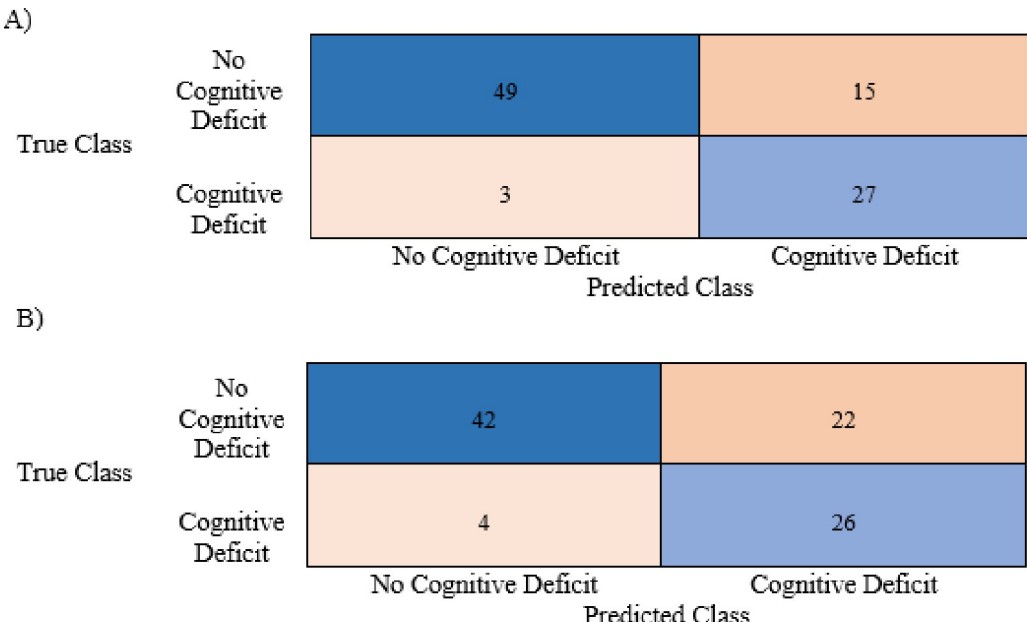

**Fig 2. Gaussian naïve Bayes confusion matrix with raw numbers.** A) With auditory variables. B) Without auditory variables.

## 3.2. Insights from the model

Shapley values were calculated to determine which predictors were the greatest contributors. *Fig 4* displays the Shapley values for the top predictors for the Gaussian naïve Bayes and kernel Naïve Bayes classifiers respectively with a greater Shapley value indicating a better predictor.

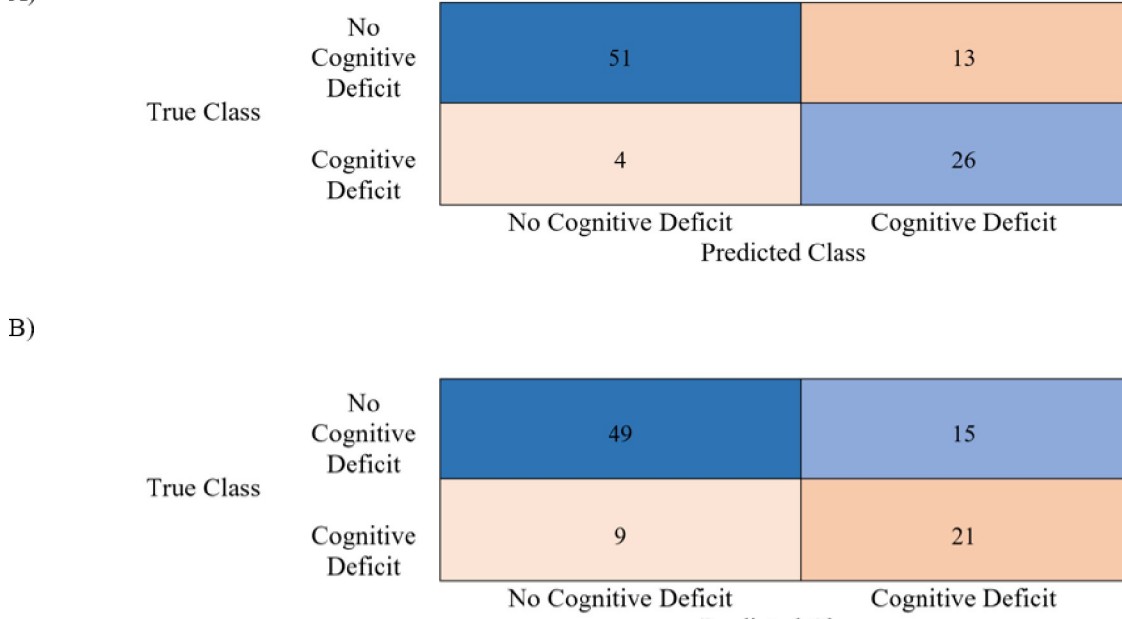

**Fig 3. Kernel naïve Bayes confusion matrix with raw numbers.** A) With auditory variables. B) Without auditory variables.

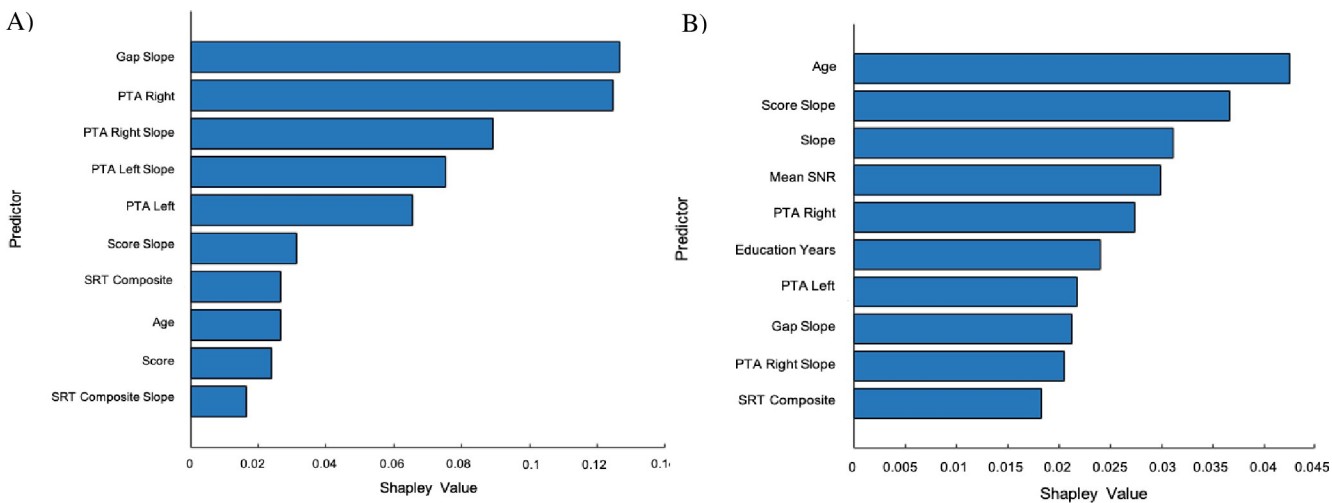

**Fig 4.** A) Gaussian Naïve Bayes Shapley values B) Kernel Naïve Bayes Shapley values.

As expected, factors such as age and education were predictors of MoCA performance. But auditory variables also contributed significantly to the models' predictive capabilities. Specifically, gap detection threshold slope, PTA values (left, right, and slope variables), SSW results, and HINT results were significant in both models. Additionally, both mean and slope variables were important predictors for the algorithms indicating that a subject's trajectory is an important factor. Interestingly, HIV status was not chosen as a significant predictor in either Gaussian and kernel naïve Bayes classifiers, but age was the most important predictor in the kernel naïve Bayes classifier.

## 4. Discussion

By using ML, we found prediction accuracy AUCs from 0.91 (*Gaussian Naïve Bayes*) to 0.87 (*Kernel Naïve Bayes)* with auditory variables included in the model. We also found model prediction to be more accurate (0.09 increase in AUC) with auditory variables included. Although a 0.09 increase in model prediction may appear modest, this is an approximate 9% increase in the model's predictive capacity. This provides evidence that auditory related variables may be used for forecasting cognitive impairment in this cohort, when combined with other variables that affect performance on neurocognitive tests. As expected, a sizable proportion of the models' accuracy depends on demographic variables, including age (Shapely value of 0.03 and 0.04 for the Gaussian and Kernel algorithms respectively) and education years (0.024 for the Kernel Naïve Bayes algorithm).

Previous results have shown a relationship between central auditory and cognitive function. This study expands on previous findings by providing more robust results using ML. ML provides a methodology to generate hypotheses about complex data patterns and confounding variables through developing predictive algorithms unachievable with classic statistical methods [29]. The results validated the hypothesis that cognitive function could be predicted better with a ML approach using central and peripheral auditory variables together with demographic variables compared to previous classical statistical methods. We believe the naïve Bayes models performed better than the other machine learning algorithms because they: 1.) are highly scalable and can handle a large number of features, 2.) have low variance and are efficient with small datasets, which means they are less prone to overfitting, and 3.) they assume independence among features, effectively ignoring correlations between them. In

other words, even if there are irrelevant features present in the dataset, Naive Bayes models can still perform well because they focus on the most discriminative features (i.e., auditory variables) that are relevant for predicting the class label (i.e., neurocognitive impairment). These findings have significant implications for understanding the interplay between auditory and cognitive domains, providing a critical starting point to define comprehensive and accurate assessments in future research and clinical applications. As we continue to utilize ML in healthcare, we anticipate further advances that will drive the advancement of personalized and targeted interventions to optimize cognitive well-being.

Peripheral auditory tests also provided value to the ML algorithms. Although all subjects had normal hearing, Shapley values indicated that PTA mean and slope values contributed to more accurate prediction from the ML algorithms. These results highlight other potentially important functional relationships of peripheral auditory input. For example, central auditory capabilities may be influenced by peripheral hearing capabilities since the pathway of auditory input may affect the brain's processing mechanisms. Additionally, the concentration necessary to perform peripheral auditory tests may involve cognitive capabilities as well as hearing ability. Studies show that peripheral auditory function relates to cognition, specifically sustained attention, and could be an independent influencing factor for older adults with more extensive hearing loss [30]. High frequency thresholds ($\geq$4 kHz) may also provide a mediator for understanding the relationship between central auditory function, age, and cognition due to their comtribution to speech understanding. These results do not diminish central auditory significance in the ML algorithms but highlight the importance in measuring and understanding the whole auditory system and its relationship to cognition.

One important practical application for a future refined and optimized ML algorithm is providing a metric of neurocognitive function for PLWH. PLWH have shown differences in brain regions and functions necessary for auditory processing including gray matter atrophy, axonal injury, loss of axonal density as well as diffuse white matter abnormalities in the internal capsule, thalamus, and corpus collosum [30, 31]. When imaging data are not available and exhaustive neurocognitive battery assessments beyond the MoCA are not feasible, central auditory tests coupled with relevant confounding variables could be analyzed via a ML algorithm to predict cognitive capabilities accurately. A clinically validated ML predictive algorithm would enable significantly more accessible testing in underprivileged areas in addition to reducing the subjectivity that can characterize cognitive tests.

While the MoCA was used in this proof-of-concept study, cognitive testing using a comprehensive neuropsychological assessment is considered the gold standard in assessing and monitoring neurocognition [32]. This form of assessment provides an estimate of global function as well as scores for domain-based skills such as language, attention, and memory. A full neuropsychological assessment, however, is costly, time consuming, and requires specialized training for interpretation. Access to such evaluations in resource limited settings, where the burden of HIV tends to be significantly higher than in developed countries, is not always feasible. Additionally, finding culturally and linguistically appropriately normed measures is challenging [33]. Also, both the MoCA and the comprehensive neurocognitive assessments are strongly affected by education and socioeconomic status.

The auditory system provides a useful tool for assessing brain function because processing auditory information is neurologically demanding but is also a naturalistic task that does not require much training or education. Speech perception, particularly interpreting speech in noise, engages several cortical and subcortical centers [34, 35]. What makes central auditory tests appealing for assessment is that they are relatively short (the Triple Digit Test takes 7 minutes), easy to explain (the Hearing-in-Noise Test and Triple Digit Test involve identifying words or numbers in background noise), do not require trained administrators, and involve

straightforward tasks that are unlikely to be strongly affected by education and socioeconomic status. These measures could be a major advance for following PLWH in the developing world. These tests, however, share some of the same limitations as neuropsychological tests in that they are affected by factors such as age, and so need to be interpreted in context, which ML allows.

While not the focus of the current study, HIV status was not selected in either the Gaussian or kernel naïve Bayes as a significant predictor or MoCA score. Clifford 2008 [36] and Spudich 2013 [37] both note that despite modern antiretroviral therapy, neurocognitive disturbances can still be detected in nearly half of HIV patients. Clifford 2017 [38] has also found that HIV may exacerbate age-associated cognitive decline and that HIV-associated neurocognitive disorders continue to occur. In this analysis, we cannot confirm this finding but show that the PLWH are more likely to show deficits on the MoCA (39.8%) compared to HIV-negative controls (26.4%). Nevertheless, the use of ML algorithms to predict cognitive function with the assistance of auditory tests as described here may help PLWH and others living in under resourced areas. These algorithms provide a direction for further research into clinically relevant applications within HIV associated neurocognitive disorder and other neurocognitive conditions. Future work that could be pursued to narrowing down the predictive features to highly predictive and easy to measure parameters.

### 4.1. Limitations

While the algorithms in this study demonstrate that ML can use auditory capabilities to predict neurocognitive dysfunction on the MoCA, a few limitations exist. The predictive feature ranking algorithms used favored linearly separable predictors, hence, this likely biased the performance of ML algorithms downstream to benefit algorithms which leverage linear relationships. Furthermore, the confidence intervals for the primary and secondary outcomes did not incorporate patient-level variation which may be possible via a non-parametric bootstrap analysis. Differneces were found in Shapley values between Gaussian naïve Bayes and the kernel naïve Bayes. With Gaussian naïve Bayes, it is assumed the data follow a Gaussian distribution. Kernel relaxes this assumption by modeling the distribution via kernel density estimation, but like the other ML models which did not perform as well, the kernel model is likely more prone to overfitting because it requires additional hyperparameters for the density function used to model the feature's distribution. Therefore, both models have costs and benefits. We were more confident in Gaussian naïve Bayes as it overperformed kernel naïve Bayes but chose to include both models to provide a comprehensive overview of data prediction. The MoCA is a cognitive screening measure and not as comprehensive as a full cognitive test battery. A ML algorithm incorporating a full neurocognitive assessment should be constructed for clinical applications. Lastly, using slope variables requires longitudinal measurements to calculate and employ the slopes. ML algorithms that do not require longitudinal data for prediction would be easier to apply. Future analyses could explore the use of other Bayesian additive regression trees and fully Bayesian approaches (i.e., Markov chain monte carlo), which are expected to outperform all frequentist methods for rare events or imbalanced data and small sample size while accounting for complex dependency structures.

### 4.2. Conclusion

The discovery that the auditory system can be a window into cognitive processes could make cognitive testing simpler and more objective. Central auditory tests in isolation, however, may not provide sufficient accuracy for predictions. This exploratory study shows that adding demographic factors as well as data from repeated measures can greatly improve the prediction

of cognitive deficits. Further development and implementation of ML algorithms using central and peripheral auditory variables for cognitive prediction could help realize the promise of these tests for prediction.

## Supporting information

**S1 File. Supplementary: A supplementary document is provided to show: 1) a summary of subject selection process for machine learning, 2) diagnostic feature ranking results using a t-test and Bhattacharyya ranking processes, 3) hyperparameter sweeps for Kernel distribution, and 4) background on Naïve Bayes algorithms.**
(DOCX)

**S2 File. Inclusivity-in-global-research questionnaire: This document outlines the ethical, cultural, and scientific considerations specific to inclusivity in global research conducted where the study took place: Dar es Salaam, Tanzania.** This questionnaire highlights the local authors at Muhimbili University of Allied and Health Sciences, Drs. Albert Magohe and Enica Massawe, that played a critical role in this study and the ongoing longitudinal study this analysis was derived.
(DOCX)

## Author Contributions

**Conceptualization:** Christopher E. Niemczak, Abigail M. Fellows, Enica R. Massawe, Jay C. Buckey.

**Data curation:** Christopher E. Niemczak, Abigail M. Fellows, Samantha M. Leigh.

**Formal analysis:** Christopher E. Niemczak, Basile Montagnese, Jiang Gui, Jay C. Buckey.

**Funding acquisition:** Albert Magohe, Enica R. Massawe, Jay C. Buckey.

**Investigation:** Christopher E. Niemczak, Jay C. Buckey.

**Methodology:** Christopher E. Niemczak, Basile Montagnese, Joshua Levy, Jiang Gui, Jay C. Buckey.

**Project administration:** Abigail M. Fellows, Samantha M. Leigh, Albert Magohe, Jay C. Buckey.

**Resources:** Enica R. Massawe.

**Software:** Joshua Levy.

**Supervision:** Christopher E. Niemczak, Joshua Levy, Jiang Gui, Samantha M. Leigh, Albert Magohe, Enica R. Massawe, Jay C. Buckey.

**Validation:** Christopher E. Niemczak, Basile Montagnese, Jiang Gui.

**Visualization:** Christopher E. Niemczak, Joshua Levy.

**Writing – original draft:** Christopher E. Niemczak, Basile Montagnese, Joshua Levy, Jay C. Buckey.

**Writing – review & editing:** Christopher E. Niemczak, Basile Montagnese, Joshua Levy, Abigail M. Fellows, Jiang Gui, Samantha M. Leigh, Albert Magohe, Enica R. Massawe, Jay C. Buckey.

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
