## [Decision Letter · Decision Letter 0]

4 Dec 2023

PONE-D-23-26242Application of Machine Learning to Predict Cognitive Deficits in HIV using Auditory and Demographic FactorsPLOS ONE

Dear Dr. Niemczak,

Thank you for submitting your manuscript to PLOS ONE. After careful consideration, we feel that it has merit but does not fully meet PLOS ONE’s publication criteria as it currently stands. Therefore, we invite you to submit a revised version of the manuscript that addresses the points raised during the review process.

We look forward to receiving your revised manuscript.

Kind regards,

Billy Morara Tsima, MD MSc

Academic Editor

PLOS ONE

2. 1) Please note that PLOS ONE has specific guidelines on code sharing for submissions in which author-generated code underpins the findings in the manuscript. In these cases, all author-generated code must be made available without restrictions upon publication of the work. Please review our guidelines at https://journals.plos.org/plosone/s/materials-and-software-sharing#loc-sharing-code and ensure that your code is shared in a way that follows best practice and facilitates reproducibility and reuse.

2) Please include a complete copy of PLOS’ questionnaire on inclusivity in global research in your revised manuscript. Our policy for research in this area aims to improve transparency in the reporting of research performed outside of researchers’ own country or community. The policy applies to researchers who have travelled to a different country to conduct research, research with Indigenous populations or their lands, and research on cultural artefacts. The questionnaire can also be requested at the journal’s discretion for any other submissions, even if these conditions are not met.  Please find more information on the policy and a link to download a blank copy of the questionnaire here: https://journals.plos.org/plosone/s/best-practices-in-research-reporting. Please upload a completed version of your questionnaire as Supporting Information when you resubmit your manuscript.

 [YES. This study was funded by the National Institutes of Health (NIH), grant number 5R01DC009972 to principal investigator Jay C. Buckey M.D. The content of this report is solely the responsibility of the authors and does not necessarily represent the official views of the NIH.].  

7. Please upload a copy of Figure S1 and S3, to which you refer in your text on page 1 and 2 in Supplementary. If the figure is no longer to be included as part of the submission please remove all reference to it within the text.

Reviewers' comments:

Reviewer's Responses to Questions

**Comments to the Author**

1. Is the manuscript technically sound, and do the data support the conclusions?

Reviewer #1: No

Reviewer #2: Yes

Reviewer #3: Yes

2. Has the statistical analysis been performed appropriately and rigorously? 

Reviewer #1: No

Reviewer #2: Yes

Reviewer #3: No

3. Have the authors made all data underlying the findings in their manuscript fully available?

Reviewer #1: No

Reviewer #2: No

Reviewer #3: No

4. Is the manuscript presented in an intelligible fashion and written in standard English?

Reviewer #1: Yes

Reviewer #2: Yes

Reviewer #3: Yes

5. Review Comments to the Author

Reviewer #1: In this work, the authors try to predict cognitive deficits in HIV using auditory measurements and demographic factors. The methodology, results and conclusions presented by the authors are unsatisfactory, incoherent and do not meet the standards of a well conducted experimentation and study.

This work is highly incremental i.e applying stock machine learning techniques to understand the predictive capability of auditory signals beyond demographic factors. The authors start by setting the objective of predicting neuro-cognitive deficits in patients living with HIV using auditory tests and demographic factors. However, the experiments conducted and results presented do not seem to provide any added value to the research field.

In Table 2, the authors compare the performance of different machine learning methods for their task showing very modest gains that does not demonstrate the importance of those small gains in the overall objective. Also, it is unclear how this prediction relates to predicting cognitive deficits in PLWH vs healthy populations.

In Figure 2, the shapely values for some of the variables are very different between gaussian naive bayes and kernel naive bayes methods which brings doubt in the model predictions. There has been no comment or further investigation provided on this observed phenomenon.

Overall, the manuscript provides little added value and is not suitable for publication. The results are unsatisfactory, not adequately substantiated with evidence, does not address the key questions in the objective and is incoherent in several ways described above.

Reviewer #2: Summary:

The paper uses machine learning (ML) algorithms to predict cognitive deficits with auditory and demographic data. In 5 of 7 ML algorithms under testing, auditory data help improves the prediction performance over models only based on demographic data. Naive Bayes based models are the best performing models (ROC-AUC around 0.9), based on which the most important features for cognitive deficits are identified. Both the statistical analysis and interpretations look good. Please see the comments below:

Major:

In 2.1 Data collection - Cognitive Data Collection, does the cognitive impairment indicator variable (MoCA score<26) the only response variable this study focus on? Is the "two sample" in T-test and the "classification error" in Bhattacharyya algorithm based on the binary variable derived from MoCA score (<26)?

The data collection was during 2017-2022, which include the sars-cov-2 pandemic period. Can the authors comments on if the data collection procedures before and after the pandemic was changed or not, if the visiting intervals of the subjects was also impacted? If yes, how would these factors influence the data and results? e.g. due to the pandemic, it is possible a subject has multiple visits before pandemic and 1 visit after pandemic, where the time interval of visits before pandemic are short while the one after the pandemic is much longer. The data of that after-pandemic single visit tends to have higher influence on the slope than the other visits before pandemic.

Can the authors discuss more about why Naive Bayes models are performing better than the others? And does the strong assumption of naive bayes hold in this study?

It seems the auditory variable trajectories are useful to predict cognitive function. Can the authors comment on is it interesting to predict the cognitive function trajectory instead of the final cognitive function?

Ensemble model represents a large class of models, which one is tested in this paper?

As is indicated in discussion, the HIV status is not a significant predictor for cognitive function. I think is makes more sense to not highlight HIV in title and abstract. After reading the title and abstract,

my expectation was that there would be a section talking about how auditory and demographic factors can be used to predict cognitive deficits specifically caused by HIV.

Minor:

Typo "2.63 Data Collection" -> "2.1 Data Collection"

In supplementary figure 1, it seems from N=557 to N=478, there are some additional exclusion criteria but the second paragraph of "2.1 Data Collection - Subjects" did not mention it.

In supplementary figure 1, typo "Exlusionary" -> "Exclusionary"

In supplementary figure 2, the feature names on y axis are missing.

In Table 3, can the authors describe how was the p-values was calculated?

Reviewer #3: The publication is well written; there is not really a lot of 'new research' with respect to machine learning, but this is an interesting use case.

Under "main outcomes" the "area under the curve" should specify the AUC is for the receiver operational characteristic (ROC) curve, and the other metrics should be cited too (F1 and Yourdon).

There are many references that describe problems or issues with the AUC measure, particularly for imbalanced data sets such as this one. Precision / recall curves would be a good complimentary measure too. The main reason for this issue is that the work is pretty straight forward but does show some level of utility, so gaining more insight into how well the concept might work would be useful. For this reason, I would also like to see the confusion matrices from the supplement inserted and worked into the main text as well, for this is often the best way to see the impact of selected thresholds (which translate ROC curves and PR curves into 'real world' impact).

6. PLOS authors have the option to publish the peer review history of their article (what does this mean?). If published, this will include your full peer review and any attached files.

Reviewer #1: No

Reviewer #2: No

Reviewer #3: No

---

## [Author Response · Author response to Decision Letter 0]

26 Feb 2024

Response to the Reviewers: 

We would like to thank the editor and reviewers for their comments and suggestions on our manuscript. We have made the necessary corrections, outlined below starting with the editor’s comments. 

Editor’s comments:

We note that the grant information you provided in the ‘Funding Information’ and ‘Financial Disclosure’ sections do not match. When you resubmit, please ensure that you provide the correct grant numbers for the awards you received for your study in the ‘Funding Information’ section.

- This has been done. 

In your Data Availability statement, you have not specified where the minimal data set underlying the results described in your manuscript can be found. PLOS defines a study's minimal data set as the underlying data used to reach the conclusions drawn in the manuscript and any additional data required to replicate the reported study findings in their entirety. 

- Due to the number of potentially identifiable features in the data used for the machine learning algorithm, Dartmouth College IRB and Muhimbilli University of Health and Allied Sciences (MUHAS) have asked to restrict data sharing to those with proper compliance certifications. We welcome data sharing but want to ensure a proper data use agreement is in place to handle data properly and with complete confidentiality. 

Reviewer #1: 

In this work, the authors try to predict cognitive deficits in HIV using auditory measurements and demographic factors. The methodology, results and conclusions presented by the authors are unsatisfactory, incoherent and do not meet the standards of a well conducted experimentation and study.

- The longitudinal study in Tanzania offers the rare opportunity to examine predictors of neurocognitive function. Studying auditory tests as potential predictors is new and novel. The extensive dataset we have is particularly well suited to machine learning approaches. We believe the understand the reviewer’s concerns and have addressed them in the responses below.

- While we recognize the limitations in this study, we believe our approach adds to the literature on how neurocognitive deficits can be found using auditory measures. This approach combines multiple demographic variables with novel auditory assessments to assess and predict neurocognitive deficits. If auditory tests can predict neurocognitive dysfunction using machine learning techniques, this could change how neurocognitive deficits, particularly related to HIV, are monitored and detected in low- middle-income countries. 

This work is highly incremental i.e applying stock machine learning techniques to understand the predictive capability of auditory signals beyond demographic factors. 

- We agree the work is incremental, but we believe an incremental approach is needed. Auditory test results correlate with neurocognitive deficits. Turning this correlation into a useful prediction requires incremental testing and the use of multiple variables. We believe starting with standard or stock machine learning approaches is a good first step. The machine learning techniques used in this manuscript offer interpretability, efficiency, robustness, and insights into feature importance. These benefits make them suitable for various research and clinical applications. If auditory test results can add to the predictive capability of neurocognitive deficits, it could dramatically change the landscape of neurocognitive treatment, monitoring, and detection.

The authors start by setting the objective of predicting neuro-cognitive deficits in patients living with HIV using auditory tests and demographic factors. However, the experiments conducted and results presented do not seem to provide any added value to the research field. 

- To our knowledge, a machine learning approach such as has been used here has not been previously attempted using auditory test results combined with demographic factors. The results show improved prediction when the auditory tests are added, which is a new result that has not been shown before. This finding is important because neurocognitive assessments can require long protocols and trained administrators. They are also greatly affected by education. Auditory tests are comparatively faster, can be administered by minimally trained personnel, and don’t require much education to understand. While we have not proved that auditory tests can substitute for neurocognitive assessments, this manuscript supports our understanding of this predictive relationship and moves us closer to providing accurate and accessible measures of cognitive dysfunction using readily acquired data. 

In Table 2, the authors compare the performance of different machine learning methods for their task showing very modest gains that does not demonstrate the importance of those small gains in the overall objective. Also, it is unclear how this prediction relates to predicting cognitive deficits in PLWH vs healthy populations.

- We have tried to make the importance of these gains clearer. Demographic factors alone are strong predictors of neurocognitive deficits, so any significant gains in predictive value from additional tests, as shown in our manuscript, are worthy of further examination. We have moved the confusion matrices from the supplementary materials to the body of the manuscript to show how gains in machine learning prediction with auditory variables can be important. For example, increased age is highly related to decreased working memory, processing speed, and executive function. Our auditory tests add a significant 9% to AUC values. The fact that auditory variables add predictive value beyond age, is very appealing. In addition, the F1 and Youden’s Index are also larger for predictive models with auditory variables compared to those without. 

In Figure 2, the shapely values for some of the variables are very different between gaussian naive bayes and kernel naive bayes methods which brings doubt in the model predictions. There has been no comment or further investigation provided on this observed phenomenon.

- We appreciate this comment. With Gaussian naïve Bayes we are assuming the data follow a Gaussian distribution. Kernel naïve Bayes relaxes this assumption by modeling the distribution via a kernel density estimation. This approach is likely more prone to overfitting because the kernel method requires additional hyperparameters for the density function used to model the feature's distribution. Any misspecification of the kernel parameters can impact the fit (e.g., if kernel bandwidth is too small), then the modeled distribution for that feature may not generalize. Both methods have pro’s and con’s. We wanted to be as comprehensive as possible, so we chose to include both measures even though Gaussian naïve Bayes overperformed kernel naïve Bayes. This has been clarified in the manuscript.

- 

Overall, the manuscript provides little added value and is not suitable for publication. The results are unsatisfactory, not adequately substantiated with evidence, does not address the key questions in the objective and is incoherent in several ways described above.

- We believe that many readers may not be aware that auditory variables could be used to improve predictions of neurocognitive performance and would find these results interesting. We agree the current results represent an incremental step in understanding how the auditory system can be used as a window into neurocognitive function; but they also show a new and novel approach to predicting neurocognitive deficits. 

Reviewer #2:

The paper uses machine learning (ML) algorithms to predict cognitive deficits with auditory and demographic data. In 5 of 7 ML algorithms under testing, auditory data help improves the prediction performance over models only based on demographic data. Naive Bayes based models are the best performing models (ROC-AUC around 0.9), based on which the most important features for cognitive deficits are identified. Both the statistical analysis and interpretations look good. Please see the comments below:

- Thank you for your review of our manuscript. We have addressed your comments below. 

In 2.1 Data collection - Cognitive Data Collection, does the cognitive impairment indicator variable (MoCA score<26) the only response variable this study focus on? Is the "two sample" in T-test and the "classification error" in Bhattacharyya algorithm based on the binary variable derived from MoCA score (<26)?

- Yes, the MoCA was the only response variable we focused on for this study. Due to the MoCA’s validated binary classifier (<26 = impairment), we felt confident that this response variable would indicate neurocognitive deficits and be understandable to readers due to the international usage of this tool. Yes, the t-test and classification error in Bhattacharyya algorithm are based on the binary MoCA scores. We have clarified this in the figure legend. 

The data collection was during 2017-2022, which include the sars-cov-2 pandemic period. Can the authors comment on if the data collection procedures before and after the pandemic was changed or not, if the visiting intervals of the subjects was also impacted? If yes, how would these factors influence the data and results? e.g. due to the pandemic, it is possible a subject has multiple visits before pandemic and 1 visit after pandemic, where the time interval of visits before pandemic are short while the one after the pandemic is much longer. The data of that after-pandemic single visit tends to have higher influence on the slope than the other visits before pandemic.

- These are important questions. We did not change the testing protocol except for the use of PPE during testing after the pandemic. We have published on the effects of PPE on neurocognitive tests using the Leiter-3 in pediatric patients, which showed no main effect of PPE on neurocognitive measures. The Leiter-3 involves much more operator interaction than the MoCA so we believe the MoCA results were likely not affected. The MoCA follows strict verbal instructions that are repeated the same way for every subject. 

- There was at least a 6-month break in subject testing due to the pandemic, but after resuming activities, most subjects resumed normal frequency of study visits (twice per year). Longitudinal plots of the data do not show major changes in slope related to the pandemic. 

Can the authors discuss more about why Naive Bayes models are performing better than the others? And does the strong assumption of naive bayes hold in this study?

- We believe naïve Bayes models performed better than other machine learning algorithms because they: 1.) are highly scalable and can handle a large number of features, making them suitable for our high-dimensional dataset, 2.) have low variance and are efficient with small datasets, which means they are less prone to overfitting, especially with our testing data of 96 data points, and 3.) they assume independence among features, effectively ignoring correlations between them. Naive Bayes classifiers assume that features are conditionally independent given the class label. This means that the presence or absence of one feature does not affect the probability of another feature occurring, given the class. In other words, even if there are irrelevant features present in the dataset, Naive Bayes models can still perform well because they focus on the most discriminative features (i.e., auditory variables) that are relevant for predicting the class label. Additionally, when the assumption is violated, naïve Bayes may give less weight to redundant features, which while affecting their feature importance could also reduce overreliance on a feature that could otherwise degrade subsequent predictive performance on a test set, relating to overfitting.

- We have added a summary of why naive Bayes models are performing better than the others in the discussion. 

It seems the auditory variable trajectories are useful to predict cognitive function. Can the authors comment on is it interesting to predict the cognitive function trajectory instead of the final cognitive function?

- This is an excellent point and predicting the trajectory of cognitive function is a long-term goal. As a first step, we first wanted to see if auditory variables (mean and slope) would predict MoCA deficits at the highest likelihood timepoint of finding impaired subjects (the last visit). That is, with increased time, more subjects will show cognitive deficits, therefore the latest visit would likely have the worst neurocognitive outcome. In a practical scenario, such as clinical settings with limited resources or time constraints, focusing on one visit also provides an approach to assessing cognitive function without the need for extensive longitudinal data collection and analysis, especially when immediate decisions need to be made.

- Nevertheless, predicting the trajectory of neurocognitive function would provide a dynamic understanding of decline over time, which can be valuable for early detection and intervention. We plan to analyze trajectories on other neurocognitive tests. With this future work we may be able to identify patterns of change, such as accelerating decline or stabilization, which may offer insights into the underlying mechanisms of cognitive impairment. This study aimed to focus on if auditory variables could add to predictive ability of neurocognitive function at one timepoint.

- We have added this to the limitations section of the manuscript. 

Ensemble model represents a large class of models, which one is tested in this paper?

- We used a LogitBoost model, which is an ensemble aggregation algorithm used primarily for binary classification. Because the ensemble aggregation method was a boosting algorithm, classification trees that allow a maximum of 10 splits composed the ensemble. One hundred trees composed the ensemble. We have added “LogitBoost” to the methods section, but because LogitBoost belongs to the overarching family of ensemble models, we chose to keep that verbiage throughout the manuscript. 

As is indicated in discussion, the HIV status is not a significant predictor for cognitive function. I think is makes more sense to not highlight HIV in title and abstract. After reading the title and abstract,

my expectation was that there would be a section talking about how auditory and demographic factors can be used to predict cognitive deficits specifically caused by HIV.

- The authors debated this same question. While the PLWH group had a higher percentage of cognitive impairment, it was not a significant predictor in any of the algorithms. We have removed HIV from the title and tempered our focus on HIV in favor or prediction of neurocognitive impairment. 

Typo "2.63 Data Collection" -> "2.1 Data Collection"

- Corrected 

In supplementary figure 1, it seems from N=557 to N=478, there are some additional exclusion criteria but the second paragraph of "2.1 Data Collection - Subjects" did not mention it.

- We corrected this in the supplementary file and in the body of the manuscript. We separated out those with hearing loss (excluded n=37) and those with abnormal middle ear function (excluded n=79). 

In supplementary figure 1, typo "Exlusionary" -> "Exclusionary"

- Corrected

In supplementary figure 2, the feature names on y axis are missing.

- Added feature names.

In Table 3, can the authors describe how was the p-values was calculated?

- In Table 3, p-values were calculated using a two-proportion z-test for accuracy, and the DeLong method (which compares two AUROC curves) for AUC. We have added this to the results section. 

Reviewer #3:

The publication is well written; there is not really a lot of 'new research' with respect to machine learning, but this is an interesting use case.

- We appreciate this comment. We believe this is an interesting application of machine learning approaches to help understand how auditory measures could predict neurocognitive dysfunction. 

Under "main outcomes" the "area under the curve" should specify the AUC is for the receiver operational characteristic (ROC) curve, and the other metrics should be cited too (F1 and Yourdon).

Main

---

## [Decision Letter · Decision Letter 1]

16 Apr 2024

Machine Learning for Predicting Cognitive Deficits using Auditory and Demographic Factors

PONE-D-23-26242R1

Dear Dr. Niemczak,

We’re pleased to inform you that your manuscript has been judged scientifically suitable for publication and will be formally accepted for publication once it meets all outstanding technical requirements.

Kind regards,

Billy Morara Tsima, MD MSc

Academic Editor

PLOS ONE

Additional Editor Comments (optional):

Reviewers' comments:

Reviewer's Responses to Questions

**Comments to the Author**

1. If the authors have adequately addressed your comments raised in a previous round of review and you feel that this manuscript is now acceptable for publication, you may indicate that here to bypass the “Comments to the Author” section, enter your conflict of interest statement in the “Confidential to Editor” section, and submit your "Accept" recommendation.

Reviewer #2: All comments have been addressed

Reviewer #3: All comments have been addressed

2. Is the manuscript technically sound, and do the data support the conclusions?

Reviewer #2: Yes

Reviewer #3: (No Response)

3. Has the statistical analysis been performed appropriately and rigorously? 

Reviewer #2: Yes

Reviewer #3: (No Response)

4. Have the authors made all data underlying the findings in their manuscript fully available?

Reviewer #2: No

Reviewer #3: (No Response)

5. Is the manuscript presented in an intelligible fashion and written in standard English?

Reviewer #2: Yes

Reviewer #3: (No Response)

6. Review Comments to the Author

Reviewer #2: The revisions look good. All the questions have been answered well. I recommend accepting it for publication.

Reviewer #3: (No Response)

7. PLOS authors have the option to publish the peer review history of their article (what does this mean?). If published, this will include your full peer review and any attached files.

Reviewer #2: No

Reviewer #3: No

---

## [Editor Report · Acceptance letter]

29 Apr 2024

PONE-D-23-26242R1 

PLOS ONE

Dear Dr. Niemczak, 

I'm pleased to inform you that your manuscript has been deemed suitable for publication in PLOS ONE. Congratulations! Your manuscript is now being handed over to our production team.

Kind regards, 

on behalf of

Dr. Billy Morara Tsima 

Academic Editor

PLOS ONE